# DENSE-TO-SPARSE GATE FOR MIXTURE-OF-EXPERTS

## ABSTRACT

Mixture-of-experts (MoE) is becoming popular due to its success in improving the model quality, especially in Transformers. By routing tokens with a sparse gate to a few experts that each only contains part of the full model, MoE keeps the model size unchanged and significantly reduces per-token computation, which effectively scales neural networks. However, we found that the current approach of jointly training experts and the sparse gate introduces a negative impact on model accuracy, diminishing the efficiency of expensive large-scale model training. In this work, we proposed `Dense-To-Sparse` gate (DTS-Gate) for MoE training. Specifically, instead of using a permanent sparse gate, DTS-Gate begins as a dense gate that routes tokens to all experts, then gradually and adaptively becomes sparser while routes to fewer experts. MoE with DTS-Gate naturally decouples the training of experts and the sparse gate by training all experts at first and then learning the sparse gate. Experiments show that compared with the state-of-the-art Switch-Gate in GPT-MoE(1.5B) model with OpenWebText dataset(40GB), DTS-Gate can obtain 2.0x speed-up to reach the same validation perplexity, as well as higher FLOPs-efficiency of a 1.42x speed-up. Our code is available [1].

## 1 INTRODUCTION

Transformer models have achieved a huge success in improving model quality via scaling data and model sizes (Brown et al., 2020; Dosovitskiy et al., 2021; Ying et al., 2021). Specifically, Kaplan et al. (2020) explored the scaling law of transformer models which shows that the model quality scales as a power-law with data sizes, model sizes and the computation. However, with the rapid increasing of the model sizes, it is hard to further scale the model to extremely large size due to the limited computation power of available hardware devices. To address these challenges, sparsely-gated Mixture-of-Experts (MoE), a popular form of conditional computation, is proposed to further increase the model size while without increasing the computational cost (e.g., FLOPS) proportionally (Bengio et al., 2013; Shazeer et al., 2017; Fedus et al., 2021; Lepikhin et al., 2021; Lewis et al., 2021b; Roller et al., 2021). Specifically, tokens are routed by a sparse gate to corresponding experts that only contain part of the full model for processing, leading to lower computational costs compared to a dense model with the same model size.

The routing strategy defined by the gate is key to MoE models, as it not only determines the model sparse structure (i.e., which experts tokens are routed to), but also affects the specialization of experts (i.e., how each expert is trained) (Roller et al., 2021). State-of-the-art approaches (Fedus et al., 2021; Lepikhin et al., 2021) usually adopt a pre-defined sparse gate (e.g., Top-1 or Top-2), and then initialize and train both the gate functions and experts jointly. However, such a *join-training over pre-defined gates* could significantly limit both the training efficiency and model quality. Particularly, at the beginning of training a MoE model, both the gate and the experts are randomly initialized. In this case, the gate does not have evidence to decide which expert to process a token, and the experts also do not have experiences to process a randomly-assigned input token. Joint-updating their models requires a vast amount of trial and errors to converge. Furthermore, the pre-defined gates significantly limit the opportunity of exploring all experts to only 1 or 2 at a time, and have to rely on some random routing noises to get feedback from other experts and finally reinforce their future routing choices. Our observation shows that such random routing in the initial stage and long-distance reinforce-based model updating in existing approaches could affect both the training time and final model quality.

---

[1] https://anonymous.4open.science/r/MoE-3D0D/README.moe.md

In this work, to overcome the limitations in existing approaches, we advocate a new training mechanism to decide the sparse gate gradually for MoE models, which is named as `Dense-to-Sparse` gate (DTS-gate). Instead of pre-defining the sparse gate in the previous works, DTS-gate starts with a dense gate that routes tokens to all the experts so that each expert gets trained sufficiently at the beginning. Then, after a short warming-up training for all experts, the gate adaptively learns the weights of routing to each expert and gradually routes tokens to fewer experts, making the training structure sparser and continuously reducing the computation cost, while keeping the model quality improving as usual. In particular, to implement the DTS-gate, our idea is to carefully control the temperature of a softmax-based routing function, so that to adjust the weights distribution among experts and control the sparsity of the MoE layer during training. In short, DTS-gate advances in two aspects. First, compared to the *joint-training of gate and experts* from scratch, DTS-gate provides a mechanism of *training gate after experts*, which could reduce a lot of random error-trails at the beginning. Second, compared to the reinforce-based model updating, starting with a dense gate allows us to get training feedbacks from all experts and adjust the routing weights directly to the right direction, which not only speedups the convergence of the gate, but also benefits for the experts specialization.

Experiments show that compared with the state-of-the-art Switch-Gate in GPT-MoE(1.5B) model with OpenWebText dataset(40GB), DTS-Gate can obtain 2.0x speed-up to reach the same validation perplexity (Figure 3(a)), as well as higher FLOPs-efficiency of a 1.42x speed-up (Figure 3(b)). Experiments also verify the ability of DST-Gate for scaling models with more experts or more MoE layers.

## 2    RELATED WORK

**Static Sparse Neural Networks**    Exploiting the sparsity in deep neural networks can reduce both storage and computation requirements for model training and inference. One of the most widely used sparsification methods is weight pruning (LeCun et al., 1990; Han et al., 2015). Previous studies propose to prune away redundant or less useful weights based on various pruning criteria (e.g., the importance of individual weights (Han et al., 2015) or groups of weights (Wen et al., 2016; Luo et al., 2017; He et al., 2017)) and then fine-tune remained weights to regain the lost accuracy. After pruning and fine-tuning, parts of weights are permanently removed, inducing a static sparsity pattern in DNNs. The sparsity pattern/structure is a trade-off between model effectiveness and hardware efficiency (Mao et al., 2017). Early works attempt to increase the sparsity ratio or model accuracy by employing unstructured sparsification methods, while recent works focus more on structured sparsity for practical speedup on hardware. Interestingly, Frankle & Carbin (2018) points out training a sparse network from scratch is superior or comparable to pruning-based methods. While Liu et al. (2018) draw a different conclusion under different settings (e.g., training hyper-parameters, pruning structures). Therefore, it still needs further investigation to answer which one is critical to sparse neural networks, the sparsity structure or remained weights. Our DTS-gate is analogous to pruning-based methods that training all experts first and then learning the sparse gate routing.

**Conditional Computation with MoE**    Different from static sparse neural networks that permanently remove some weights, conditional computation (Bengio et al., 2013) activates only some relevant parts of the model on a per-example basis, which can be regarded as a dynamic sparsity structure that remains all weights but brings sparsity into the computation. The MoE model (Shazeer et al., 2017), as a specific form of conditional computation, contains many experts and a trainable gating network which selects a sparse combination of experts to process each input sample. Conditional computation is capable of reducing inference cost (without reducing model capacity) or increasing model capacity (without increasing inference cost) from a model acceleration or scaling perspective. On the other hand, the activated parts (experts in MoE) can be regarded as structured sparse blocks, which does not introduce additional computational overhead. However, conditional computation models are often difficult to train, since they require learning discrete routing decisions from individual examples to experts and the gating network tends to converge to a state that only selects the same few experts (Eigen et al., 2013). Shazeer et al. (2017), Lepikhin et al. (2021) and Fedus et al. (2021) add auxiliary load-balancing losses to mitigate this self-reinforcement phenomenon and improve training efficiency. In such MoE models, the gating network and experts, as two critical components, are jointly trained which may interfere with each other. In DTS-gate, we consider

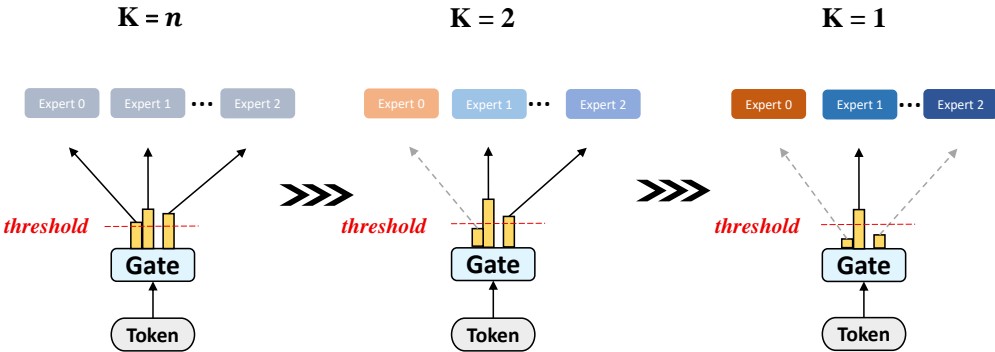

Figure 1: Overview of training an MoE with DTS-Gate. At the beginning, the experts are not trained and tokens would be distributed to all experts to enable sufficient updates for experts. As the training continues, the output weights of gate begin to diverge because experts tend to have their own domain. The gate becomes sparser. Finally, the weights are close to one-hot and each token is distributed to the most suitable expert.

decoupling the training of experts and the gating network by involving all experts starting with a high temperature in Gumbel-Softmax and then training the gating network to be sparser and select the best expert through decaying this temperature. Lewis et al. (2021b) formulates token-expert allocation as a linear assignment problem and guarantees balanced compute loads. Roller et al. (2021) replaces the gating network with a hash-based routing strategy, e.g. random hash, clustered hash dispersed hash.

## 3 METHODS

### 3.1 PRELIMINARY

The main components of a sparse MoE layer are a network $G$ as sparse gate and an expert network $E$.

**Sparse Gate Network** The sparse gate network $G$ takes a token $x_s$ ($x_s \in \mathbb{R}^D$) as input and produces the probability of it with respective to N experts. Equation 1 formulates the gate function, where $W_g (W_g \in \mathbb{R}^{D \times N})$ is the embedding of N experts and the top-K outputs are normalized via $softmax$ distribution. The sparse gate network is widely adopted by Shazeer et al. (2017); Fedus et al. (2021); Lepikhin et al. (2021) and achieves promising results with constant computation cost.

$$g(x_s) = softmax(TopK(x_s \cdot W_g)) \tag{1}$$

**Expert Network** The expert network $E$ is a set of N experts $\{e_1, ..., e_N\}$, where each is a *FFN* layer and contains its own parameters in our models. For each expert $e_i$ ($e_i : \mathbb{R}^D \to \mathbb{R}^D$), it takes the token $x_s$ as input to produce its own output $e_i(x_s)$. The final output of the expert network $y_s$ is the linearly weighted combination of each expert's output on the token by the gate's output, formulated as Equation 2.

$$y_s = \sum_{i=1}^{N} g(x_s)_i e_i(x_s) \tag{2}$$

### 3.2 DENSE-TO-SPARSE GATE

Sparse gating has demonstrated its superior model efficiency in both training and inference, but prior work tends to convergence to a sub-optimal model on FLOPs-efficiency or sample-efficiency because of the jointly training of the randomly initialized gate network and expert network. In

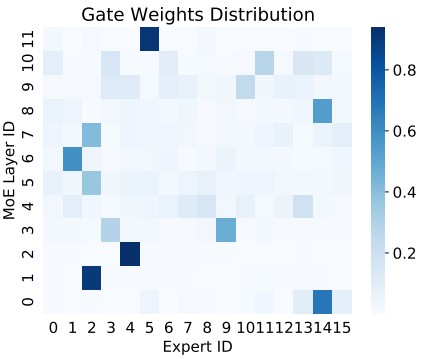
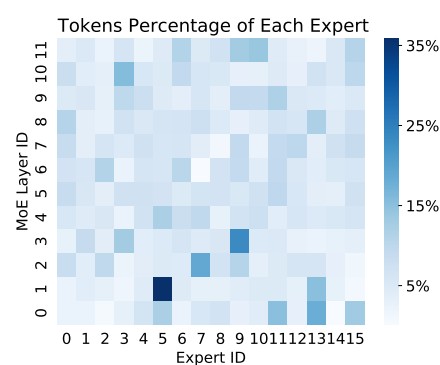

(a) Weights Distribution of a Single Token      (b) Loads Distribution within a Batch (40960 tokens)

Figure 2: The 16 experts' weights and loads distribution of GPT-MoE-Dense (all experts are activated for each token) among 12 MoE-layers (totally 24 layer). Figure 2(a) shows that the gate's weights to each expert is uneven. And in Figure 2(b), we route tokens to its Top-1 expert based on the DenseMoE gate, and the loads between experts is naturally imbalanced.

this paper, we propose a new mechanism for the gate function, named `Dense-to-Sparse` gate (DTS-Gate), which starts as a dense gate that routes tokens to all experts and then gradually becomes sparser. DTS-Gate benefits from the sufficient training of experts in the early stage and then make the experts selection becomes sparser on the basis of specialized experts. This dense-to-sparse process only occupies a small fraction compared with the total training time, which usually takes days to weeks.

**Sparsity in Mixture of Experts**  Sparsity naturally exists in MoE models. Here we train a 24-layer transformer decoder model, with every other FFN layer replaced by 16-expert MoE layer. Each MoE layer is densely connected, where tokens are distributed to all experts but with weighted differently. As shown in Figure 2, the expert weight matrix is extremely sparse and the loads are naturally imbalanced among experts.

**Dynamic Sparsity Gate**  In order to control the sparsity during training, we adopt the softmax temperature to adjust the weights distribution among experts. Formulated as Equation 3, $W_g$ is the experts embedding, $\zeta$ is the extra noise and sampled from Gumbel$(0, 1)$ distribution (Jang et al., 2017), and $\tau$ is the softmax temperature which controls the distribution. When the $\tau$ increases, the distribution becomes more uniform, which evolves more experts into the computation of each token. As the $\tau$ approaching 0, the distribution becomes one-hot, which is more confident for the gate network.

$$g(x_s) = \frac{e^{(x_s \cdot W_g + \zeta)/\tau}}{\sum_{s'=1}^{N} e^{(x_{s'} \cdot W_g + \zeta)/\tau}} \tag{3}$$

**Sparsity Scheduler**  By scheduling the temperature of Equation 3, we can control the sparsity of the MoE layer at different training stages. There is a trade-off between small temperature, where experts distributions are nearly one-hot and lead to large variance of gradients in learning but less FLOPs in computation. Large temperature is just the opposite. DTS-Gate starts at a large $\tau$ and anneal to a small $\tau$ with linearly decaying.

**Adaptive Capacity**  With $\tau$ decreasing, the weights among experts approach one-hot distribution, where most of them are close to zero. To optimize such sparsity into actucal training time, we use threshold drop the experts whose weights fall below the threshold as Figure 1, and no extra communication or computation will be wasted. Furthermore, we don't enforce a balanced assignment algorithm for DTS-Gate because we observe naturally imbalanced distribution of experts from

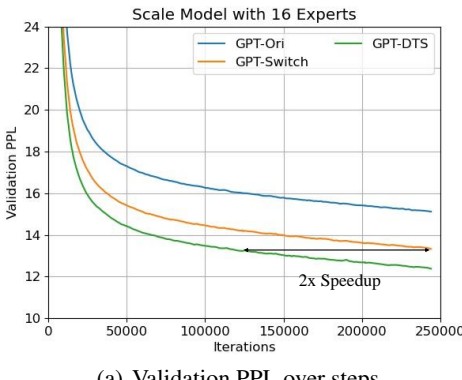
(a) Validation PPL over steps

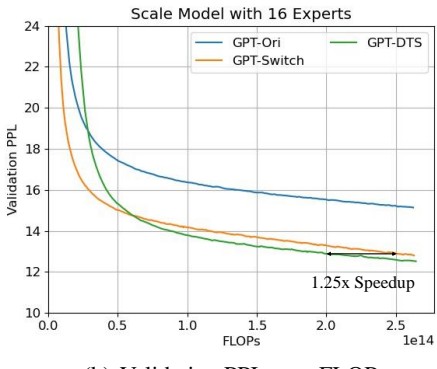
(b) Validation PPL over FLOPs

Figure 3: End-to-end performance comparison between three models. Figure 3(a) and Figure 3(b) represent the curve of PPL over iterations and FLOPs, where DTS-Gate can obtain 2.0x speed-up to reach the same validation perplexity, as well as higher FLOPs-efficiency of a 1.42x speed-up.

Table 1: Model Architectures

| $Model$ | $Params$ | $Layers$ | $d_{Model}$ | $d_{FFN}$ | $N_{MoE}$ | $N_{Experts}$ |
|---|---|---|---|---|---|---|
| GPT-Small | 117M | 12 | 768 | 3072 | — | — |
| GPT-Medium | 345M | 24 | 1024 | 4096 | — | — |
| GPT-MoE | 1.5B | 24 | 1024 | 4096 | 12 | 16 |

Dense-MoE models, shown as Figure 2(b). To meet the demand of above two designs, we enable each expert with adaptive capacity to make them well specialized.

## 4 EXPERIMENTS

### 4.1 EXPERIMENTAL SETUP

**Dataset**   Models are trained on two datasets separately: English Wikipedia (17.5GB) (Devlin et al., 2019) and WebText (40GB) (Radford et al.). Wikipedia is a single domain of text, which contains about 2,500 words, and WebText is a large and diverse dataset in various domains, which is collected using 45 million links from Common Crawl. We apply GPT2-BPE (vocabulary size of 50257) on both datasets for tokenization and binarize the dataset for loading efficiently.

**Evaluation**   Our experiments are conducted on language modeling task, and perplexity (PPL) is reported as the evaluation metric of model performance. We focus exclusively on comparing FLOPs-efficiency, defined as the best model performance (PPL) can be achieved given the fixed number of floating-point operations. Meanwhile, we don't add the FLOPs of balance loss throughout experiments in this paper, as we focus on gate distribution with or without balance loss. We don't report actual training time because the actual training time mixes the computation complexity with system implementation and leads to unfair comparisons as new algorithms without well optimized system implementation. In the appendix, we will show the routing pattern of DTS is relative stable, with heavy-loaded experts almost keep fixed. And we can utilize system techniques, e.g., partitioning hot experts among multiple GPUs and merging multiple cold experts onto less GPUs, for load balance between GPUs.

**Model Architectures**   We test on three models, including GPT-Small, GPT-Medium and GPT-MoE, and the detailed architectures are shown in Table 1, where GPT-Small and GPT-Medium are two standard architectures defined by Radford et al.. We scale the *feed-forward* layer (FFN) of GPT-

Medium(345M) from 1 to 16 as the MoE Layer, and replace every other FFN as MoE to match the total parameters of GPT-Large(1.5B). We also extend GPT-Small(117M) with various MoE-layer settings to evaluate our DTS-Gate method.

**Baselines**    As for MoE-based models, we compare 4 methods totally, including Switch(Top-1 gate proposed by Fedus et al. (2021)), DTS-Dynamic(decay the temperature from 2 to 0.3 linearly in the first 15000 steps, and switch to Top1 at the 20000 step), and Dense-MoE(select all the experts for each token). The rank of theoretical FLOPs per token is GPT-MoE-Dense > DTS > Switch = GPT-Small(excluding the cost of gate network and all_to_all communication in MoE layer).

**Training Hyperparameters**    All models use the same hyperparameters of 4,000 warmup steps, polynomial learning rate scheduler, Adam optimizer, clip norm of 5.0, weight decay of 0.1 and dropout of 0.1. We use GeLU activation functions (Hendrycks & Gimpel, 2016) and set sequence length as 1024. We use CrossEntropy as the criterion and 0.01 as the coefficient of balance loss in Switch. To save the training cost, we use batch size of 256 and sequence length of 768 on the GPT-Small model, other hyper-parameters keep constant. We set the threshold of gates as 0.001 across the training, which is a hyper-parameter to determine how large the expert's weight is important and is a trade-off between computation and accuracy from our point of view.

## 4.2    RESULTS AND ANALYSIS

### 4.2.1    END TO END COMPARISON

To improve the computation efficiency, only part parameters are used for each token in sparse models with the cost of model performance. DTS-Gate aims to shift the model training from dense to sparse, and keep the inference cost same as sparse models, and results are shown in Figure 3. Experiments show that compared with the state-of-the-art Switch-Gate in GPT-MoE(1.5B) model with OpenWebText dataset(40GB), DTS-Gate can obtain 2.0x speed-up to reach the same validation perplexity (Figure 3(a)), as well as higher FLOPs-efficiency of a 1.42x speed-up (Figure 3(b)). Experiments also verify the ability of DST-Gate for scaling models with more experts or more MoE layers.

**Comparison with Dense Models**    Dense models, where all parameters are participated in computation for each token, tends to achieve high model performance due to its large parameter capacity but with high computation cost. As shown by Figure 3(a), MoE-Dense performs better than other models at the same iteration. By utilizing the dense-to-sparse training mechanism, MoE-DTS shrinks the performance gap between MoE-Dense and MoE-Switch. As for FLOPs-efficiency, MoE-DTS outperforms both dense models because of its sparse computation.

**Comparison with Sparse Models**    MoE-Switch pre-defines its static Top-1 gating network and jointly training the gate and experts networks. Different from MoE-Switch, MoE-DTS utilizes temperature to adjust the distribution of the token-to-experts (one-hot or uniform) and threshold to remove computation of experts with low weights. MoE-DTS performs better than MoE-Switch in sample-efficiency because of more experts involved in training and updates at beginning, shown as Figure 3(a). As for FLOPs-efficiency, DTS-Gate first involve more experts into warm-up training, which is poor FLOPs-efficency. But with the training going on, DTS-Gate can obtain greater than 5% improvements in FLOPs-efficiency compared with the state-of-the-art Switch-Gate in GPT-MoE(1.5B) model with OpenWebText dataset(40GB).

It is worth noting that several hyper-parameters of MoE-DTS, such as max/min temperature and decay iterations, needs to be carefully determined. In this paper, we just propose the training mechanism of Dense-To-Sparse and prove its effectiveness through a simple policy.

### 4.2.2    SCALABILITY

In this section, we conduct experiments about the scalability and FLOPs-efficiency of DTS-Gate. Although it doesn't adopt a balanced routing algorithm, experiment results show that DTS-Gate can be utilized in scaling models, because it wouldn't converge to only a few experts .

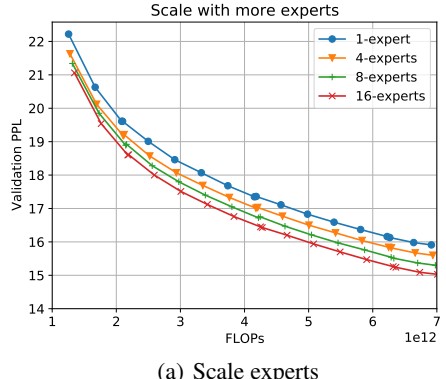
(a) Scale experts

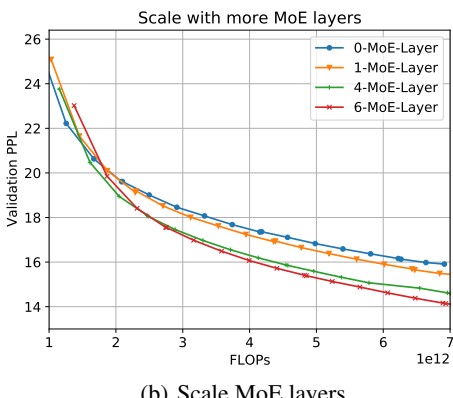
(b) Scale MoE layers

Figure 4: Scalability for MoE-DTS models. It shows that More experts or more MoE-layers (larger models with constant FLOPs), will lead to better FLOPs-efficiency.

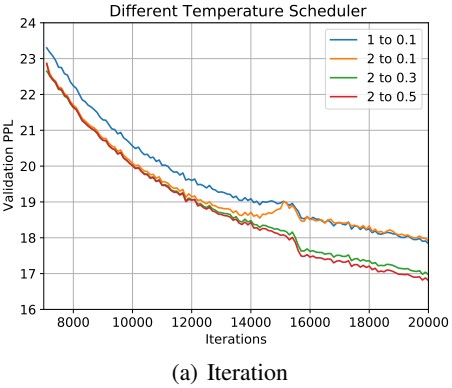
(a) Iteration

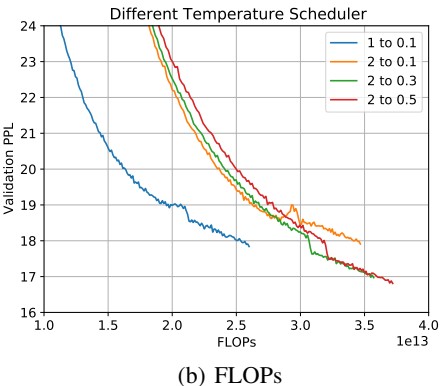
(b) FLOPs

Figure 5: Different Temperature Scheduler. We vary the temperature interval values and experiments show that larger beginning temperature will cost more computation at early stage, while provide little extra gain (1 to 0.1 V.S. 2 to 0.1). Small temperature at end will lead to the model degradation, show by 1 to 0.1 and 2 to 0.1 at 15000 iteration in Figure 5(a)

**Increase the Expert Number**    Based on the 12-layer GPT-Small model(117M), we replace the 7th FFN layer by one MoE layer and vary its experts number within $\{1, 4, 8, 16\}$. As shown by Figure 4(a), MoE-DTS keeps consistent improvements when increasing expert numbers.

**Increase the MoE layer number**    Similarly, we also vary the number of MoE layer within $\{0, 1, 4, 6\}$ and each MoE layer contains 8-experts. Figure 4(b) shows that by increasing MoE layers, MoE-DTS can achieve better model performance with same FLOPs.

### 4.2.3    EFFECT OF SPARSITY SCHEDULER

Our work proposes the mechanism of Dense-To-Sparse training for MoE models, and employ Gumbel-Softmax to sparsify the MoE layers by decreasing the temperature $\tau$. In this subsection, several experiments are conducted to show the effect of different temperature schedulers($\tau-$scheduler). The used models are GPT-MoE, 24-layer decoder with 12 MoE-layer (16 experts) and dataset is OpenWebText(40GB).

We decay $\tau$ from max_value to min_value in the first 15000 iterations and switch to Top1 at the 20000 step. Experiments with different max_value and min_value are evaluated and the results are shown in Figure 5.

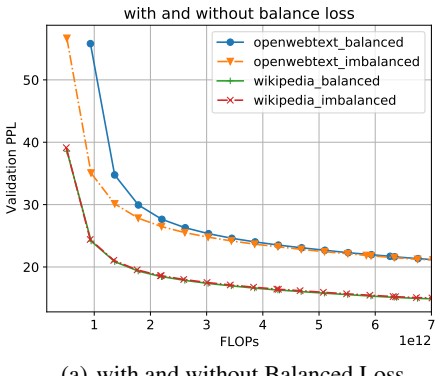
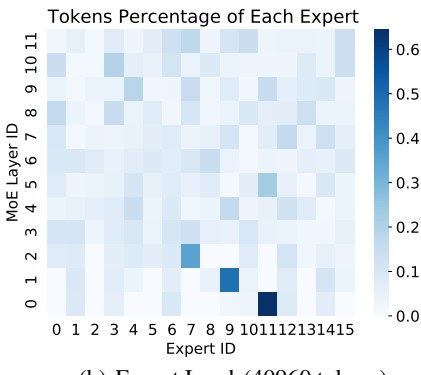

(a) with and without Balanced Loss  (b) Expert Loads(40960 tokens)

Figure 6: Effect of Balanced Loads

**Max/Min Temperature**   Under small temperatures, the weight distribution of experts is close to one-hot, which leads to the one-token-one-expert distribution and high computation efficiency, but the variance of gradients is large. In the contrast, large temperatures result in nearly uniform distribution gate weights, which evolves more experts into training but the variance of gradients is small. We schedule the temperature $\tau$ to start from a large value and anneal it to a small $\tau$ gradually. Figure 5 shows the sample-efficiency and FLOPs-efficiency of different $\tau-$ schedulers.

By comparing $\{1\ to\ 0.1\}$ and $\{2\ to\ 0.1\}$, we find larger begin temperatures will lead to high sample-efficiency but low FLOPs-efficiency, because more experts are participated in computation. By comparing $\{2\ to\ 0.1\}$ and $\{2\ to\ 0.3\}$, we find smaller end temperatures cause poor model performance because of the large variance of gradients. After analysis about these experiments, we choose $\{2\ to\ 0.3\}$ as our training configs.

**Decay Iterations**   After max/min temperatures are determined, we need to care about the decay iterations. $\{2\ to\ 0.1\}$, $\{2\ to\ 0.3\}$ and $\{2\ to\ 0.5\}$ spend about 11800, 13200 and 15000 iterations respectively for decaying temperature from 2 to 0.5. Higher decay rate performs good at sample-efficiency at the expense of FLOPs-efficiency. Choosing suitable decay rate is a balance between sample-efficiency and FLOPs-efficiency. Actually, considering about the training cost, we usually set $5\% - 15\%$ as the decay iterations.

**Dynamic V.S. Static**   Both large temperatures and small temperatures have their own advantages and disadvantages, it would be better to design a $\tau-$ scheduler according the models and hardware for the balance between model-performance and FLOPs-efficiency.

### 4.2.4   EFFECT OF BALANCED LOADS

Current works utilize either extra balance loss (Lepikhin et al., 2021; Fedus et al., 2021) or balanced assignment algorithm (Lewis et al., 2021a) to avoid the training of sparse MoE-layer converge at the status of only a few experts are frequently used. Balanced loads are naturally easy for efficient system implementation. However, it is still unknown about which is better for model performance. We conduct several experiments to explore this mystery. Meanwhile, DTS-Gate proposed by this paper is orthogonal to the balance loss, because DTS-Gate cares about the weights' polarization at the token level while balance loss cares about the loads of each expert at model level/layer level.

**With and Without Balance Loss**   Balance loss can improve the performance of switch transformer Fedus et al. (2021). Therefore, we conduct experiment with and without balance loss on DTS, to study whether it can also improve the model performance of DTS. To analyze the influence of balance loss on DTS-Gate, we replace the 7th layer of GPT-Small(117M) models with one MoE layer(16experts), and training this model with and without balance loss on OpenWebText and Wikipedia separately. As shown by Figure 6(a), this model performs almost the same on Wikipedia with and without balance loss, while it performs better without balance loss on OpenWebText at the

begining and then gradually coincide with the one with balance loss. The difference may introduced by that OpenWebText is a large and diverse dataset in various domains but Wikipedia only comes from one domain. Therefore, imbalanced distribution is suitable for training on OpenWebText. But Why OpenWebText with and without balance loss are similar at the end of training. We check the experts' loads in next paragraph.

**Expert Loads Distribution**   We directly load the checkpoints from the end to end experiments, where MoE-DTS is the 1.5B MoE models described as above. Figure 6(b) shows that the experts's loads of the 3rd to 11th MoE layer is almost balanced, which is learned without balance loss.

## 5   CONCLUSION AND FUTURE WORK

MoE models suffer from the model quality challenge due to the difficulty of training the pre-defined sparse gate and the experts jointly. To address this challenge in MoE training, we presented DTS-Gate that (1) designs the dense-to-sparse training mechanism and (2) provides the scheduling policy for sparsity adjusting. Our evaluations show that DTS-Gate can not only achieve better model quality in Transformers with given computation budget but also achieve better FLOPs-efficiency when comparing with previous works in MoE training. DTS-Gate is a stepping stone in exploring MoE training. On the other hand, DTS-Gate opens challenges for system execution due to the computation in the early stage and the adaptive capacity of experts. In the future, we would like to design and implement system-level optimizations to achieve efficient training in both model quality and system execution.

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

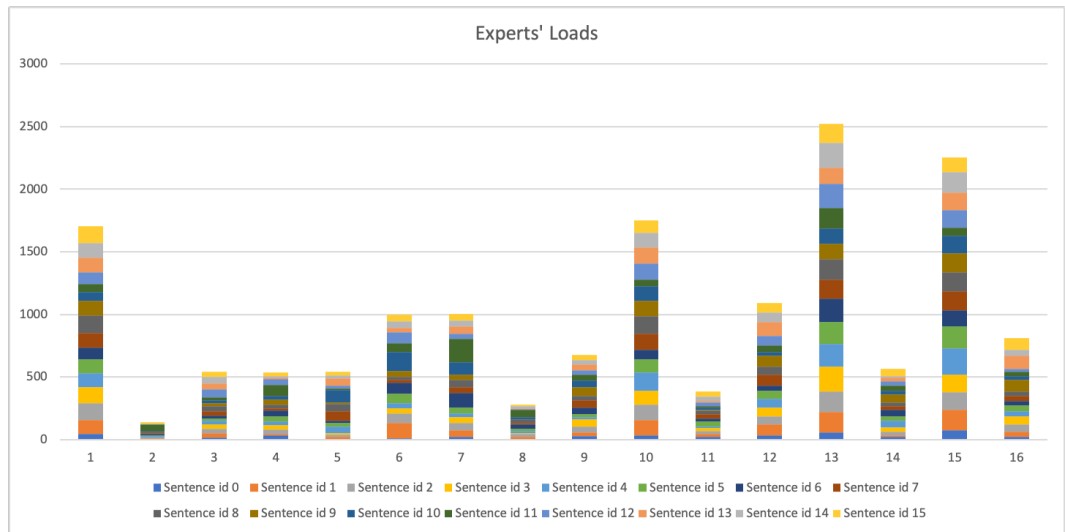

Figure 7: The Loads of each expert in $7th$ MoE-Layer, with 16 experts in total, accumulated of 16 sentence.

Noam Shazeer, Azalia Mirhoseini, Krzysztof Maziarz, Andy Davis, Quoc V. Le, Geoffrey E. Hinton, and Jeff Dean. Outrageously large neural networks: The sparsely-gated mixture-of-experts layer. In *5th International Conference on Learning Representations, ICLR 2017, Toulon, France, April 24-26, 2017, Conference Track Proceedings*. OpenReview.net, 2017. URL `https://openreview.net/forum?id=B1ckMDqlg`.

Wei Wen, Chunpeng Wu, Yandan Wang, Yiran Chen, and Hai Li. Learning structured sparsity in deep neural networks. *Advances in neural information processing systems*, 29:2074–2082, 2016.

Chengxuan Ying, Tianle Cai, Shengjie Luo, Shuxin Zheng, Guolin Ke, Di He, Yanming Shen, and Tie-Yan Liu. Do transformers really perform bad for graph representation? *arXiv preprint arXiv:2106.05234*, 2021.

# A APPENDIX

## A.1 EXPERT LOADS IN DTS

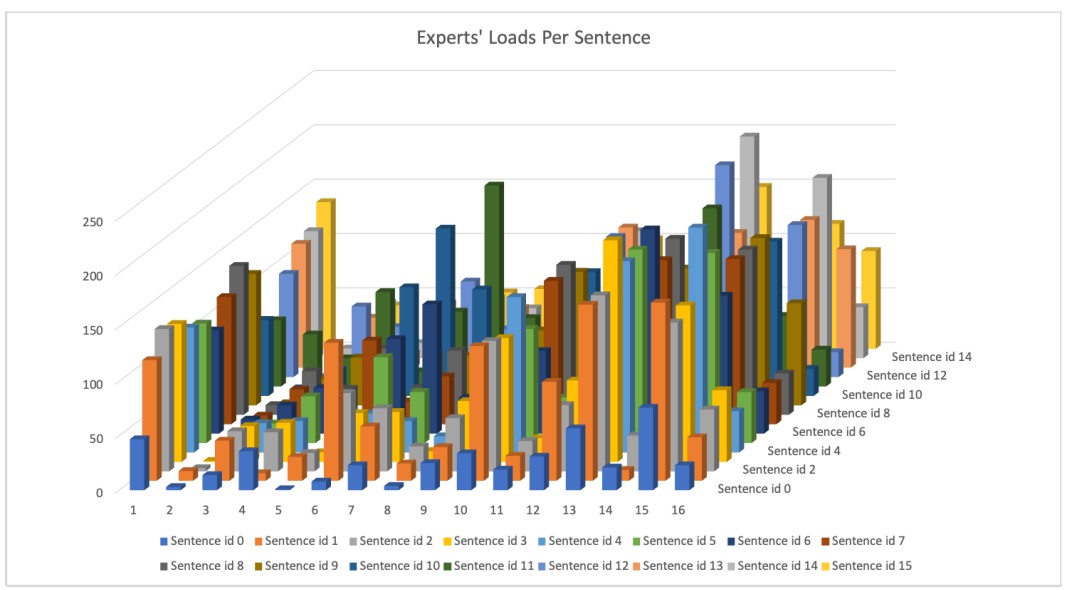

Figure 8: The Loads of each expert in $7th$ MoE-Layer, with 16 experts in total, in the sentence level.

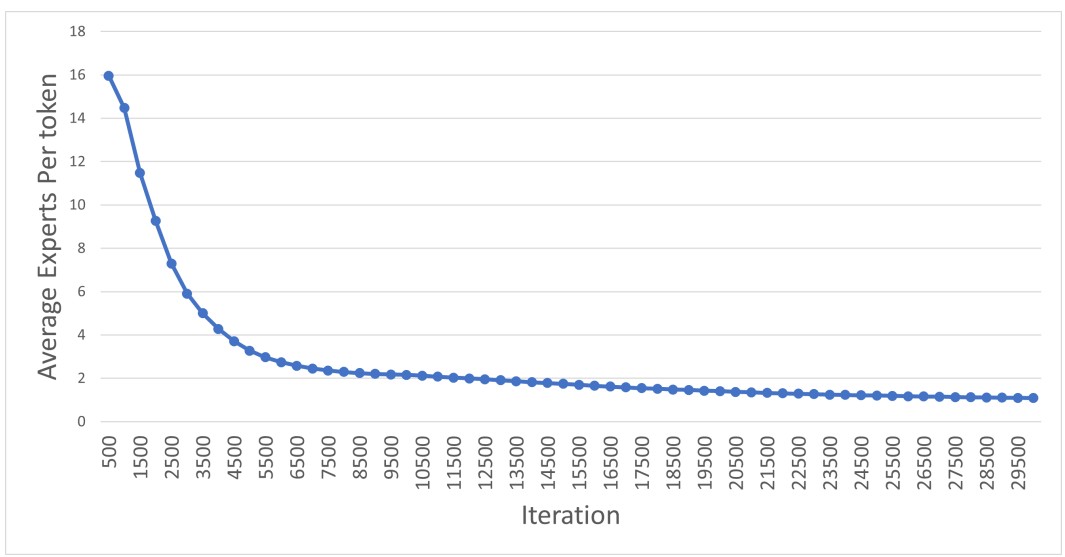

Figure 9: Average expert per token during training, with 16 experts in each MoE layer.

