# OpenReview forum: "Dense-to-Sparse Gate for Mixture-of-Experts"
_ICLR.cc/2022/Conference — ICLR 2022 Submitted_

### Official Review · Reviewer_hQQS · 2021-11-03

**Correctness:** 2
**Technical Novelty And Significance:** 2
**Empirical Novelty And Significance:** 2
**Recommendation:** 5
**Confidence:** 3

**Main Review:**

**Strong Points:**
- using gradual sparsification for the MoE gates of a transformer model seems to offer a final performance (perplexity in this particular case) close to the MoE dense gates, while having lower computational requirements. At least in theory, as I believe that a binary mask is used to emulate sparsity. Is my understanding correct?

**Weak Points and suggestions:**
- The paper claims are not perfectly aligned with the empirical evaluation. If the proposed gradual sparsification with DST is intended to be for any MoE model than other neural network models besides transformers shall be studied. If the proposed method is particularly designed for transformer models then perhaps vision transformers shall be also studied. Based on the chosen direction, new algorithmic baselines and datasets have to be added for comparison. Otherwise, the broadness of the claims needs to be adjusted accordingly.
- The related work discussion is missing very important work. Particularly, gradual sparsification for dense-to-sparse training has been introduced in [1], up to my best knowledge. From that moment, a large body of works has been released on this topic. In parallel, sparse-to-sparse training has been studied for vision transformer in [2]. Please add a consistent paragraph to discuss the above directions and to highlight clearly what is novel in this paper in comparison with those works. Otherwise, the proposed method, broadly speaking, would be just a simple application of [1] (or follow ups) on MoE gates for transformers. A recent survey can give more details on these topics [3].
- Can you please add an algorithm to better illustrate the proposed method?
- Is the lower bound from the “Adaptive Capacity” paragraph fixed across training? How can one choose it?
- Why in Figure 3b, the dense model minimum PPL is higher than in Figure 3a? Can you train all models from Figure 3a for the same number of iterations? Now, somehow, it seems that the complete overview picture is missing
- I suggest to carefully proof read the whole paper. The English usage and general appearance can be improved. Also, there are a number of typos, e.g., “Comparsion with…”. “…As shown in Figure 3.2….”

**References:**
- [1] Christos Louizos, Max Welling, Diederik P Kingma. Learning Sparse Neural Networks through 𝐿_0 Regularization. ICLR 2018, https://openreview.net/forum?id=H1Y8hhg0b
- [2] Tianlong Chen , Yu Cheng , Zhe Gan , Lu Yuan , Lei Zhang , Zhangyang Wang. Chasing Sparsity in Vision Transformers: An End-to-End Exploration. NeurIPS 2021, https://arxiv.org/abs/2106.04533
- [3] Torsten Hoefler, Dan Alistarh, Tal Ben-Nun, Nikoli Dryden, Alexandra Peste. Sparsity in Deep Learning: Pruning and growth for efficient inference and training in neural networks, JMLR 2021, https://www.jmlr.org/papers/v22/21-0366.html


**Summary Of The Paper:**

This paper introduces the idea of using dense-to-sparse (DTS) training gates in mixture-of-experts (MoE) based on a gradual sparsification process. The novel produced method is evaluated on a Transformer model. Overall, it looks to me that the paper has rather limited novelty. While the paper is well-written, in my opinion, the empirical evaluation support is not perfectly aligned with the paper claims.

**Summary Of The Review:**

While there are some clear advantages highlighted by this paper in obtaining more efficient transformer models, overall, I believe that the paper is not ready yet for publication.

---

> ### Author Response · Authors · 2021-11-23
> **Response to Reviewer 5**
>
> **Q1: The end2end comparison experiment.**
>
> Sorry for the confusing experiment results. After we fix some implementations and relax the hard Top-1 into soft at the end of warm-up stage, which means we allow tokens to select Top-2 or more experts if their weights are larger than the threshold, DTS performs better than Switch significantly, shown as Figure3. We find it brings many gains in model performance at the cost of a few extra computations because of the relaxed condition, e.g., 1.063 experts per token on average (slightly greater than 1).
> We have updated our new experiments in this paper (Figure 3), and the results show our proposed DTS achieves significant gains over Switch. Specifically, compared with GPT-Switch, DTS is more sample-efficient and yields a 2x speed-up to reach the same validation perplexity (Figure 3a), as well as more FLOPs-efficient and a 1.25x speed-up (Figure 3b). Experiments are conducted on OpenWebText datasets (40GB), with the GPT-MoE model architecture (24-layer MoE Transformer, 16 experts for every other layer, 1.5B parameters in total), which is the same parameter size and dataset as the largest GPT-2.
>
> **Q2: The claims are not perfectly aligned with the empirical evaluation.**
>
> Thanks for pointing out this concern. Our proposed method is designed for transformer based MoE models. The experiments on Vision MoE models are our ongoing work. We will update these results to paper in the future, because of the large computation resource demand.
>
> **Q3. Missing related work’s discussion. And highlight the novel in comparison with them.**
>
> Thanks for pointing out this concern. We will update it in the related work section.
> This paper proposes DTS, a simple yet effective training mechanism, which activates experts from densely to sparsely for MoE model. Although this mechanism is intuitive, we explore the routing pattern of various transformer settings during training and find the gate distribution changes a lot in the early training stage and almost keeps stable later. Based on this observation, we propose the dense-to-sparse mechanism, which aims to learn a better gate with extra computation in the early stage.
> The related works are about multi-task learning and try to learn the parameter sharing pattern among different tasks, which aims to solve the model performance degradation when same layers shared between unrelated tasks. Our paper aims to relieve the differentiable operation of TopK selection among experts based on the gates warm-up in the early stage (from dense to sparse).
>
> **Q4. Add an algorithm to illustrate proposed algorithm.**
>
> Thanks for your insightful suggestion. We will add this algorithm to the paper in the future.
>
> **Q5: The lower bound from “Adaptive Capacity”:**
>
> Thanks for pointing out this missing detail. We set the lower bound threshold as 0.001 across the training. It is a hyper-parameter to determine how large the expert's weight is important and we think it is a trade-off between computation and accuracy. If it is smaller, then more computation will be involved while higher model performance will be achieved.
>
> **Q6. Carefully Proofread.**
>
> We will carefully proofread the paper and make it more intuitive.

---

> > ### Comment · Reviewer_hQQS · 2021-11-29
> > **Thanks for your answers.**
> >
> > I thank the authors for considering and answering to my comments.
> >
> > While their answers clarified some aspects, I see that there are still some places for improvements in the paper: e.g., the experiments on Vision MoE - please note that I really understand the unfortunate limited resources reality; a part of the related work issue that starts with "Particularly, gradual sparsification..." from my review; and so on. I really appreciate the fact that the authors acknowledge these issues and mention in their answers that they will address them in the next version of the paper. However, all these changes, when put together, may slightly change the main message of the paper and, therefore, the paper may need a new peer-review process. Corroborating this with the relatively low level of novelty from the actual revision, I still believe that the paper is not ready yet for publication. Thus, I will keep my initial recommendation and I recommend to the authors to incorporate all the received feedback in the next revision.

---

### Official Review · Reviewer_XPcr · 2021-11-03

**Correctness:** 2
**Technical Novelty And Significance:** 2
**Empirical Novelty And Significance:** 2
**Recommendation:** 5
**Confidence:** 4

**Main Review:**

This paper tackles an interesting aspect of conditional computation: in the context of MoEs, how do we start training from scratch when individual experts can easily break down and collapse?

The paper provides an intuitive answer: we can start by training all experts together (i.e. as a dense model), and relax this over time, hopefully, once all individual experts are enough developed. In some setups (which, interestingly, coincide with some of the original motivation ones for MoEs) this may not be possible, for example if the number of experts is just huge.

Some recent work has shown the effect of increasing "k" a lot (see [1], Figure 10). It seems that the effect of increasing "k" plateaus somewhat quickly. While in those experiments "k" doesn't change over time, I think this may suggest there's little to gain in doing so (especially as it's quite expensive).

Overall, I think the experiments in the paper do not show the proposed algorithm offers any advantage over standard top-k routing (Switch, in the paper).

Figure 3 seems to be the only comparison between Switch ([2] i.e. no warm-up) and DTS-Gate (expert warm-up). First, the reported metric is training perplexity. I'm not sure why this is the case: test (or validation) perplexity should be reported. In the second plot (3b), which is the most relevant in my opinion (FLOPs or runtime, rather than "steps" as in 3a, as these hide very different costs), it seems that Switch dominates DTS. For any FLOPs budget (vertical line), Switch offers better performance than DTS. Similarly, for any attainable performance level, Switch gets there at a lower FLOPs cost. Thus, I'm not sure how the paper justifies the use of DTS. I may have misunderstood something though. It's not clear to me why the DTS run (in red) was halted, and didn't continue training. There's a mismatch between both plots for the yellow curve (MoE-Dense), in the left one (3a), perplexity is below 14, while in the right one (3b), it's above 15. Why? Was the x-axis (FLOPs) trimmed?

In the previous to last paragraph in page 6, there's this sentence: "DTS-Gate can obtain greater than 5% improvements in FLOPs-efficiency compared with state-of-the-art Switch-Gate [...]". Where can this be seen?

Also, and as the authors acknowledge in the conclusions, the fact that DTS doesn't impose balance (if I understand correctly) even when using a load balancing loss (as opposed to having a maximum expert capacity that enforces it), can lead to an inefficient use of current hardware (some experts waiting for others).

For Figure 6, as expert id's are independent across layers, it would be nicer to sort experts per row according to their load, so that clearer patterns emerge.

Alternative idea: One could try something even simpler. Initially train a dense model (with one MLP only, rather than E different MLPs + router) for a number of steps. Then, replicate the dense MLP to all the experts, and add a router from scratch, and apply top-k as usual. In other words, pre-train a dense backbone for a bit, and then initialize the expert model from this. Have the authors tried this? I can see this working well.


**Typos:**

Second paragraph of page 6: "set tokens of each sample as 1024" -> "set tokens' hidden size to be 1024"?

"comparsion" --> "comparison"


**References:**

[1] Scaling Vision with Sparse Mixture of Experts

[2] Switch Transformers: Scaling to Trillion Parameter Models with Simple and Efficient Sparsity

**Summary Of The Paper:**

The paper proposes an algorithm, DTS-Gate, to warm-up experts in an MoE learning setup. Rather than applying standard top-k selection per input, the algorithm initially applies dense routing (i.e. k = total number of experts), to avoid quick collapse, and gradually relaxes it to become sparser (and cheaper). Experiments and ablations in language tasks are provided to support the advantages of the algorithm.

**Summary Of The Review:**

While the algorithm is reasonable, the experimental results (only one plot, Figure 3b) do not suggest it provides any additional benefit compare to the standard baseline.

------------

Note: After the rebuttal, I increased the score from 3 to 5.

---

> ### Author Response · Authors · 2021-11-23
> **Response to Reviewer 4**
>
> Thanks for your constructive feedback! We believe that addressing this feedback will make our paper better.
>
> **Q1: The end2end comparison experiment.**
>
> Sorry for the confusing experiment results. After we fix some implementations and relax the hard Top-1 into soft at the end of warm-up stage, which means we allow tokens to select Top-2 or more experts if their weights are larger than the threshold, DTS performs better than Switch significantly, shown as Figure3. We find it brings many gains in model performance at the cost of a few extra computations because of the relaxed condition, e.g., 1.063 experts per token on average (slightly greater than 1).
> We have updated our new experiments in this paper (Figure 3), and the results show our proposed DTS achieves significant gains over Switch. Specifically, compared with GPT-Switch, DTS is more sample-efficient and yields a 2x speed-up to reach the same validation perplexity (Figure 3a), as well as more FLOPs-efficient and a 1.25x speed-up (Figure 3b). Experiments are conducted on OpenWebText datasets (40GB), with the GPT-MoE model architecture (24-layer MoE Transformer, 16 experts for every other layer, 1.5B parameters in total), which is the same parameter size and dataset as the largest GPT-2.
>
> **Q2: Imbalanced workloads (No expert capacity) will lead to inefficient use of hardware.**
>
> Thanks for your concerns! Because the actual training time mixes the computation complexity with system implementation and leads to unfair comparisons as new algorithms without well optimized system implementation. Detailed analysis (Appendix) shows that the routing pattern of DTS is relative stable, with heavy-loaded experts almost keep fixed. And we can utilize system techniques, e.g., partitioning hot experts among multiple GPUs and merging multiple cold experts onto less GPUs, for load balance between GPUs. Furthermore, the system-level optimization about how to deal with dynamic capacity of different experts is our ongoing work.
>
>
> **Q3. DTS cannot be utilized to scale a huge number of experts.**
>
> In some setups (which, interestingly, coincide with some of the original motivation ones for MoEs) this may not be possible, for example if the number of experts is just huge.
> Thanks for pointing out this concern. Although it is hard to train all the experts at the beginning when the number of experts is huge, but the number of activated expert K anneals quickly. The change of K during training is shown in (Appendix).
>
> **Q4: The effect of increasing "k".**
>
> Thanks for pointing out this concern. We agree with you about increasing "k" plateaus quickly. But the suitable “k” is different for different models on different datasets and tasks. To make this problem simpler, we begin as the dense (k=all the experts) and quickly anneal it to a suitable K by using temperature and threshold.
>
> **Q5: Experiment results on validation datasets:**
>
> Sorry for causing this concern. We have updated all the experiment figures with validation results through all the paper.
>
> **Q6: Sort experts per row in Figure 6:**
>
> Thanks for your suggestions!
>
> **Q7: Alternative idea: Initially train a dense model and then replicate it for multiple experts:**
>
> Thanks for your suggestions! In the early attempt of MoE, we have an implementation like your suggestion, where we set the same random seed for different experts. And the results are worse than both naïve switch and DTS.

---

> > ### Comment · Reviewer_XPcr · 2021-11-30
> >
> > Thank you for your detailed response.
> >
> > I acknowledge that the new Figure 3 conveys a significantly different outcome with respect to the original plot (more aligned with the claims made in the paper). Also, the models seem to have been trained for a significantly longer amount of FLOPs/time (after the initial submission deadline?). I raise my score from 3 to 5.
> >
> > Still, overall, I'm not convinced this is a practical idea for large MoE models with many experts (due to the fact that we start by training a "dense" model at the beginning), and I think the work needs a bit more polishing.

---

### Official Review · Reviewer_wm4x · 2021-11-04

**Correctness:** 4
**Technical Novelty And Significance:** 4
**Empirical Novelty And Significance:** 3
**Recommendation:** 6
**Confidence:** 4

**Main Review:**

Notes:
* The idea is an interesting one. If the quality of sparse models can be significantly improved through using more compute only early on in training, this could become a widely adopted technique.
* Reasonably well-written paper.
* The authors use a reasonable baseline of a 24-layer MoE Transformer, 16-experts for every other layer.
* Visualization of the model dynamics, specifically gate weights distribution, is a nice addition beyond plots and tables of numbers.

Improvements:
* I suggest a different graphic for the “experts” in Figure 1. Our field already anthropomorphizes neural networks enough; I’d prefer to not see a little human with an etched brain as a stand-in for two matrix multiplications and a non-linearity.
* A style recommendation. Make each of your captions self-contained with description and result so that the reader can quickly scan your work figure-to-figure. Elaborate over the short phrases you’re using (e.g. “Figure 5: Different Temperature Scheduler”).
* On page 8, don’t use the abbreviations w/ and w/o.
* The significance of the gate values in Figure 7 will likely be lost to many readers. As stated above, please elaborate in your captions to describe what is being shown and why it’s important to substantiate your contribution.
* p8. “Though these tokens are spatial proximity” -> Fix.

Questions:
* Figure 3. Why are there periodic dips in the training perplexity curves? If this is from repeating the data in the same order, I’d recommend in the future that you shuffle data each epoch.
* Figure 4 demonstrates that MoE-DTS improves with more experts and expert-layers. How does this scaling compare to MoE-Switch?
* “experiments results show that the FLOPs-efficiency can be optimized into time-efficiency” -> Where was this shown? All the plots and data I see are for FLOPs, not wallclock time on a specific hardware.
* “It is amazing that DTS-Gate can learn to load balance without extra loss.” -> Was this verified to be a property only of DTS-Gate? Please confirm that naive approaches without load balancing do not learn this, or else, perhaps remove this. Also, as a nit, I’d avoid subjective claims like in using the word “amazing”.

**Summary Of The Paper:**

This work proposes the Dense-to-Sparse Gate or DTS-Gate. It’s a simple idea of starting training with soft (continuous) routing to all experts and then gradually reducing to hard (discrete) routing. The authors claim that this improves the quality of traditional approaches. The work proposes the novel idea and details how to schedule the sparsity adjustment. The authors suggest this improves traditional approaches by 5% FLOPs-efficiency.

**Summary Of The Review:**

I’m positively disposed towards this work, however, my main concern is on the lack of a clear demonstration of the magnitude and the significance of the gains.

Please clearly show me the asymptotic quality is worse using vanilla approaches (e.g. Lepikhin et al., 2020, Fedus et al., 2021). As far as I can tell, the only substantiating plot for this is 3(b), but in my opinion, this doesn’t clearly showcase the improvement of the method. At the very least, these curves should be extended until MoE-DTS is conclusively better than MoE-Switch on a FLOPs basis. Further, though the idea is interesting, showing stronger empirical results than a 5% FLOPs-efficiency gain is likely needed to convince practitioners and researchers. If this is the case, I will improve from a 6->8.

---

> ### Author Response · Authors · 2021-11-23
> **Response to Reviewer 3.**
>
> We appreciate your assessment of the proposed DTS algorithm "If the quality of sparse models can be significantly improved through using more compute only early on in training, this could become a widely adopted technique".
> Thanks for your constructive feedback and the insightful suggestions! We believe that addressing this feedback will make our paper significantly better.
>
> **Q1: The end2end comparison experiment.**
> Sorry for the confusing experiment results. After we fix some implementations and relax the hard Top-1 into soft at the end of warm-up stage, which means we allow tokens to select Top-2 or more experts if their weights are larger than the threshold, DTS performs better than Switch significantly, shown as Figure3. We find it brings significant model performance gains at the cost of a few extra computations because of the relaxed condition, e.g., 1.063 experts per token on average (slightly greater than 1).
> We have updated our new experiments in this paper (Figure 3), and the results show our proposed DTS significantly outperforms Switch. Specifically, compared with GPT-Switch, DTS is more sample-efficient and yields a 2x speed-up to reach the same validation perplexity (Figure 3a), as well as more FLOPs-efficient and a 1.25x speed-up (Figure 3b). Experiments are conducted on OpenWebText datasets (40GB), with the GPT-MoE model architecture (24-layer MoE Transformer, 16 experts for every other layer, 1.5B parameters in total), which is the same parameter size and dataset as the largest GPT-2.
>
>
> **Q2: The FLOPs-efficiency as for time-efficiency.**
> Thanks for your concerns! Because the actual training time mixes the computation complexity with system implementation and leads to unfair comparisons as new algorithms without well optimized system implementation. Detailed analysis (Appendix xxx) shows that the routing pattern of DTS is relative stable, with heavy-loaded experts almost keep fixed. And we can utilize system techniques, e.g., partitioning hot experts among multiple GPUs and merging multiple cold experts onto less GPUs, for load balance between GPUs. Furthermore, the system-level optimization about how to deal with dynamic capacity of different experts is our ongoing work.
>
> **Q3: The periodic dips in the training perplexity curves.**
> Thanks for pointing out this question and providing solutions, which solves the dips! The new experiments is updated to the paper (Figure 3) and we change the y-axis from training perplexity
> to validation perplexity
>
> **Q4: Experiments about scaling with MoE-Switch.**
>
> Thanks for this improvement, and we will update it to the paper in the future.
>
> **Q5: Naïve approaches without load balancing.**
>
> The property about DTS and Switch-No-balance loss on whether all the experts are involved in computation.
> MoE-Switch converges to a few experts in each MoE-layer, while our method DTS learn this without balance loss. From our best of knowledge, DTS, which involves all the experts in computation and leads to more efficient parameters, avoids the serious unbalanced situation for naive approaches without balance loss.  And we will update the end2end performance and detailed analysis about load balancing between DTS and switch-no-balancloss to the paper.

---

> > ### Comment · Reviewer_wm4x · 2021-11-30
> > **Rebuttal Response**
> >
> > Thank you for addressing my concerns -- I believe the original draft is improved. However, the review from prfk has made me aware of the related work from Hazimeh et al. (2021) (https://arxiv.org/abs/2106.03760).  A snippet form their paper: "At the start of training, it uses all the available experts, so conditional training is not possible. As we discuss in Section 3, after a certain point during training, DSelect-k converges to a small subset of the experts, and then conditional training becomes possible."
> >
> > This is certainly close in spirit, but differs in approach. Hazimeh et al. (2021) was only arxived on Jul 7th and the ICLR deadline was Oct 2nd, so a case could made for "concurrent" work. However, given this prior work diminishes the novelty and a reader doesn't know which approach is better, I don't feel comfortable increasing to an 8. For a future version please do a direct comparison to this to show the advantages/disadvantages.
> >
> > Given the improvements through this rebuttal process, I'm confident this will make a great resubmission at a future venue.

---

> > > ### Author Response · Authors · 2021-12-02
> > > **Thanks for your kind suggestions**
> > >
> > > Dear Reviewer wm4x:
> > >
> > > Thanks for your kind suggestions and bringing our attention to the concurrent work of DSelectK (https://arxiv.org/pdf/2106.03760.pdf).
> > >
> > > Regarding the similarity between our work DTS and DSelectK, we believe they are different essentially as they are trying to address **two totally different problems**, though both works show SparseMoE is better than DenseMoE by coincidence.
> > >
> > > For our DTS work, DenseMoE performs **well** but costly for large model pretraining, so we tried to find a more efficient solution (SparseMoE); For the DSelectK work, DenseMoE performs **badly** for multi-task learning, so they tried to find a better solution to deal with various tasks, i.e., SparseMoE also by coincidence.
> > >
> > > Furthermore, although DTS and DSelectK use same “dense-to-sparse” mechanism, the detailed strategies are different and it is **not trivial** task to transfer the DSelectK to our scenario, as the scalability issue is more critical in our case.
> > >
> > > We explain the differences in detail in the following.
> > >
> > > **1.Different Research Problems.**
> > >
> > > + **Our DTS:** The research problem we are targeting is how to achieve better model performance with FIXED computation budget. Note that the common knowledge is: DenseMoE would perform better than SparseMoE (e.g., TopK or DTS) with UNLIMITED computation budget in this pretraining scenario. However, DenseMoE yield higher computational cost inevitably [1]. To reduce the cost of DenseMoE, previous SparseMoE approaches adopt a pre-defined sparse gate (e.g., Top-1 or Top-2).
> > > To utilize the computation budget better, we observe the gate of DenseMoE and have these two key insights:
> > >     1. The token-to-expert distribution changes a lot in the early training stage and almost keeps stable later.
> > >     2. Experts‘weights to each token are uneven and sparse (Figure 2(a)).
> > >
> > >     Based on these insights, we propose the dense-to-sparse mechanism, which aims to learn a better gate with extra computation in the early stage.
> > >
> > > + **DSelectK:** The research question of DSelectK is to find better experts ‘sharing patterns among related and unrelated tasks in multi-task learning (MTL). DenseMoE would perform worse than SparseMoE, which is also shown in Table 1 (DSelectK v.s. Softmax MoE) in its paper, since sharing experts between unrelated tasks can potentially degrade performance under this MTL scenario. This is different from why we apply the dense-to-sparse mechanism in the pretraining scenario, as mentioned above.
> > >
> > > **2.Scalability Consideration.**
> > >
> > > DTS and DSelecK were proposed for problems of two different scales, with DTS having much better scalability than DSelectK. We have verified the scalability and FLOPs-efficiency of DTS on 1.5B parameters models with 176 experts.
> > >
> > > However, DSelectK has poor scalability because of two aspects:
> > > + It only contains two states, dense(N) and sparse(K), and can’t control the dense iterations relative to total iterations. For example, dense training accounts for 42.33% in Multi-MNIST and even 80.02% in Multi-Fashion (Appendix Table C.4). DTS gradually and smoothly becomes sparser (N->N-1->…->2->1), shown in Appendix Figure 9.
> > > + The DSelectK algorithm is hard to calculate as it introduces extra learned parameters z and needs to derive the $r(S(Wx))$ (Equation 4) to update these parameters. Meanwhile, the computation of it will increase with more experts.
> > >
> > > Therefore, DSelectK may suffer from **low efficiency** when scaling to more experts and more MoE layers and can’t be widely adopted in MoE models. We will include the results in the camera-ready-copy if we have chance.
> > >
> > > Reference:
> > >
> > > [1] Kaplan J, McCandlish S, Henighan T, et al. Scaling laws for neural language models[J]. arXiv 2020.

---

### Official Review · Reviewer_prfk · 2021-11-04

**Correctness:** 3
**Technical Novelty And Significance:** 2
**Empirical Novelty And Significance:** 2
**Recommendation:** 5
**Confidence:** 3

**Main Review:**

Strengths
The proposed scheme to train dense gating then making it sparser makes sense and could potentially stabilize and improve the training of gating networks.

The authors conducted extensive experiments to evaluate both the effectiveness and efficiency of the proposed algorithm.

Weaknesses.
Stabilizing the training of the sparse gating network has been the main challenge of the MoE based model architectures, there exists some research with similar ideas trying to make the training of MoE more robust.
1. This paper uses gumbel noise to stabilize the learning of softmax gating. (https://arxiv.org/abs/1910.04915)
2. This paper uses a binary gating with similar noise and a l0 regularizer to control the sparsity. (https://ojs.aaai.org/index.php/AAAI/article/view/3788)
3. This paper uses a binary encoding and diffienable operators to smooth the learning of sparse gating, with initializing to be dense and later converging to sparse. (https://arxiv.org/abs/2106.03760)

Given these existing literatures, I feel like the novelty of this paper is quite limited.

In the meantime, since the sparsity of the gating is controlled by the annealing of temperature of the softmax, only Top-1 gating is supported, this also limits the capacity flexibility of the MoE based morel architecture.


**Summary Of The Paper:**

This paper studied the problem of training the gating network for the mixture-of-experts based model architectures. To make the gating network training more stable and robust, the authors proposed a dense-to-sparse gating training algorithm that uses Gumbel noise and temperature tuning. The authors conducted experiments on large NLP datasets.

**Summary Of The Review:**

Overall, I think the training scheme from dense to sparse for sparse MoE gating makes sense, and this is supported by experiments in the paper. However, I also think the technical novelty of this paper is limited given existing work with similar ideas on improving the gating network training.

---

> ### Author Response · Authors · 2021-11-23
> **Response to Reviewer 2.**
>
> We appreciate your assessment of the proposed DTS algorithm "the training scheme from dense to sparse for sparse MoE gating makes sense". Thanks for your constructive feedback! We believe that addressing this feedback will make our paper better.
>
> **Q1: Limited Novelty compared with related work.**
> Thanks for pointing out this concern.
> This paper proposes DTS, a simple yet effective training mechanism, which activates experts from densely to sparsely for MoE model. Although this mechanism is intuitive, we explore the routing pattern of various transformer settings during training and find the gate distribution changes a lot in the early training stage and almost keeps stable later. Based on this observation, we propose the dense-to-sparse mechanism, which aims to learn a better gate with extra computation in the early stage.
> The related works are about multi-task learning and try to learn the parameter sharing pattern among different tasks, which aims to solve the model performance degradation when same layers shared between unrelated tasks. Our paper aims to relieve the differentiable operation of TopK selection among experts based on the gates warm-up in the early stage (from dense to sparse).
>
> **Q2: The annealing of temperature in softmax will limits the capacity flexibility of the MoE based model architecture.**
>
> Thanks for pointing out this concern. Our paper focuses more on extra gains for MoE models brought by the dense-to-sparse training mechanism. The choice of using Gumbel-Softmax and annealing the temperature is just a potential policy. Any other suitable policies can also be adopted by DTS.
> Moreover, Kool Wouter, et al. have already proposed “Gumbel-Top-k Trick”[1], which can be also adopted by DTS to support flexibility of MoE-based models.
>
> Reference:
> [1] Kool, Wouter, Herke Van Hoof, and Max Welling. "Stochastic beams and where to find them: The gumbel-top-k trick for sampling sequences without replacement." International Conference on Machine Learning. PMLR, 2019.

---

> > ### Comment · Reviewer_prfk · 2021-11-29
> > **Thanks for the response!**
> >
> > After carefully reading the response, I still feel like the novelty of this work is limited.
> >
> > While training from densely activated experts first and sparsely activated experts later can be robust, existing work has been applying similar approaches. I think even though some of the related work has a multitask application, the purpose as well as their dense-to-sparse  mechanism can be easily adopt to NLP as well as other single task learning applications using MoE.

---

> > > ### Author Response · Authors · 2021-12-02
> > > **Thanks for your reply!**
> > >
> > > Dear Reviewer prfk:
> > > Thanks for your continuous efforts in replying to our responses.
> > >
> > > We agree with you that multitask MoE work (i.e., DSelectK, the one you mentioned https://arxiv.org/pdf/2106.03760.pdf) also applied dense-to-sparse mechanism to address multitask problem, however the purpose is **exactly different**.
> > >
> > > For our DTS work, because DenseMoE performs **well** but cost for large model pretraining, we tried to find a more efficient solution (SparseMoE). While for the multi-task work, because DenseMoE performs **badly** for multi-task learning, they tried to find a better solution to deal with various tasks, i.e., DSelectK. Therefore, the two works have clearly different motivations.
> > >
> > > Furthermore, although DTS and DSelectK use same “dense-to-sparse” mechanism by coincidence, the detailed strategies are different and it is **not trivial** task to transfer the DSelectK to our scenario, as the scalability issue is more critical in our case.
> > >
> > > We explain these differences in detail in the following.
> > >
> > > **1. Different motivations.**
> > >
> > > + **Our DTS:** We are targeting to achieve better model performance with FIXED computation budget. The common knowledge is: DenseMoE would perform better than SparseMoE (e.g., TopK or DTS) with UNLIMITED computation budget in this pretraining scenario. However, DenseMoE yield higher computational cost inevitably [1].
> > > Unlike previous SparseMoE approaches adopting a pre-defined sparse gate (e.g., Top-1 or Top-2), we observe the gate of DenseMoE and have these two key insights:
> > >   1. The token-to-expert distribution changes a lot in the early training stage and almost keeps stable later.
> > >   2. Experts‘weights to each token are uneven and sparse (Figure 2(a)).
> > >
> > >   Based on these insights, we propose the dense-to-sparse mechanism to learn a better gate with extra computation in the early stage.
> > >
> > > + **DSelectK:** It is targeting to find better experts sharing patterns among related and unrelated tasks in multi-task learning (MTL). DenseMoE would perform worse than SparseMoE, which is also shown in Table 1 (DSelectK v.s. Softmax MoE) in its paper, since sharing experts between unrelated tasks can potentially degrade performance under this MTL scenario. This is fundamentally different from why we apply the dense-to-sparse mechanism in the pretraining scenario, as mentioned above.
> > >
> > > **2. Scalability Consideration.**
> > > DTS and DSelecK were proposed for problems of two different scales, with DTS having much better scalability than DSelectK. We have verified the scalability and FLOPs-efficiency of DTS on 1.5B parameters models with 176 experts.
> > >
> > > However, DSelectK has poor scalability because of two aspects:
> > >   1. It only contains two states, dense(N) and sparse(K), and can’t control the dense iterations relative to total iterations. For example, dense training accounts for 42.33% in Multi-MNIST and even 80.02% in Multi-Fashion (Appendix Table C.4). DTS gradually and smoothly becomes sparser (N->N-1->…->2->1), shown in Appendix Figure 9.
> > >   2. The DSelectK algorithm is hard to calculate as it introduces extra learned parameters z and needs to derive the $r(S(Wx))$ (Equation 4) to update these parameters. Meanwhile, the computation of it will increase with more experts.
> > >
> > > Therefore, DSelectK may suffer from **low efficiency** when scaling to more experts and more MoE layers and can’t be widely adopted in MoE models. We will include the results in the camera-ready-copy if we have chance.
> > >
> > > Reference:
> > > [1] Kaplan J, McCandlish S, Henighan T, et al. Scaling laws for neural language models[J]. arXiv 2020.

---

### Official Review · Reviewer_gUQG · 2021-11-09

**Correctness:** 3
**Technical Novelty And Significance:** 2
**Empirical Novelty And Significance:** 2
**Recommendation:** 5
**Confidence:** 5

**Main Review:**

The paper proposes a simple and easy to implement idea. However, the experiments do not clearly show that Dense-To-Sparse (DTS) is better in terms of accuracy-compute trade-offs than fully sparse baselines (such as Switch-Moe).

In particular, although Figure 3b should show this (FLOP vs. Accuracy), but the lowest perplexity is achieved by Switch-MoE with $8 \cdot 10^{33}$ FLOP and DTS-MoE only matches Switch-MoE at around $6.5 \cdot 10^{13}$ FLOP. Most importantly, FLOP is not very well correlated with actual training time (nor energy consumption), since if some sort of "masking" is used to implement DTS, rather than true sparsity, some compute is wasted to compute the experts outputs that are then masked.

To this reviewer, it is not clear if masking or some thresholding was used to evaluate DTS. Figure 1 depicts a "threshold", which could mean that DTS avoid computing the output of some experts (the ones whose weight falls below the threshold), and that the FLOPs depicted in Figure 3b and thorugh the rest of the paper are more correlated with actual runtime. However, this "threshold" is not mentioned anywhere else in the paper, so it's very likely that when the temperature is very low (i.e. only 1 of the expert's weights is effectively non-zero) all the experts are being actually used. In any case, notice that even if the reported FLOPs correlate well with runtime, the observation regarding Switch-MoE is still true.

Moreover, many of the experiments do not report results on a test set, not even a validation set. Figures 3 (arguably the main plot) and 5 report *training* perplexity, rather than the perplexity on a separate validation or test set. However, Figure 4 and 6 do report results on the validation set. It's unclear to this reviewer what's the criterion used to report results on one set or the other. In any case, validation perplexity, or even better test perplexity, should have been used through all the paper (training measuraments are relevant sometimes, but not in the context of this paper).

There are some results from the "effect of balanced loads" analysis (Section 4.2.4) which are also not clear to this reviewer, as well. In particular, Figure 6a shows that the use of balanced loads adds an overhead of approximately $5 \cdot 10^{11}$ FLOPs in the openwebtext dataset. Why? Why only in this task but not in the Wikipedia task? In addition, notice that one of the main practical reasons to use a balanced load (e.g. in Switch-MoE) is to efficiently use all available compute, which is not considered in the analysis presented.

Minor typos and comments:

- Abstract: "**an** neegative impact" -> "**a** negative impact".
- Page 4: "Here we train a 24-layer transformer decoder model, with **and** every other FFN" -> "Here we train a 24-layer transformer decoder model, with every other FFN".
- Section 4.2.3: There seems to be notes left from the draft of the paper. "Need We analyze it in token-level's weights?".
- Section 4.2.4: "We directly load**s**" -> "We directly load".

**Summary Of The Paper:**

In the context of neural networks with Mixture of Experts (MoE) layers, the paper proposes to decrease the number of activated experts per each input as training progresses, by means of decreasing the temperature in the softmax that is typically used to weight the contribution of the different experts in the MoE layers. The proposed approach is compared with fully Dense MoEs and the recent Switch-MoE baselines in a language modeling task.

**Summary Of The Review:**

The main claim of the paper (the current approach of jointly training experts and the sparse gates introduce a negative impact on model accuracy) is not well supported by the experiments. In addition to that, many of the experiments report training measurements rather than test or validation measurements. Comparing training perplexity is not enough to show the superiority of the proposed approach. Some of the details of the implementation are not clear to this reviewer and affect the FLOP vs. Accuracy comparison. Actual runtime comparison (on a particual hardware architecture) would be much better.

Given all these issues, I recommend to not accept the paper.


**Update after rebuttal**
The authors have addressed many of my concerns during the rebuttal period satisfactorily. My concern about the claimed "speed-up" is still present, but the authors have stated why they decided to use FLOPs rather than actual runtime on a given hardware and implementation. Based on all this, I'm slightly increasing my score for this submission.

---

> ### Author Response · Authors · 2021-11-23
> **Response to Reviewer 1. Part I**
>
> Sorry for the confusing experiment results and thanks for your constructive feedback! We believe that addressing this issue will make our paper better.
>
> **Q1: The end2end comparison experiment**
>
> Sorry for the confusing experiment results. After we fix some implementations and relax the hard Top-1 into soft at the end of warm-up stage, which means we allow tokens to select Top-2 or more experts if their weights are larger than the threshold, DTS performs better than Switch significantly, shown as Figure3. We find it brings many gains in model performance at the cost of a few extra computations because of the relaxed condition, e.g., 1.063 experts per token on average (slightly greater than 1).
> We have updated our new experiments in this paper (Figure 3), and the results show our proposed DTS achieves significant gains over Switch. Specifically, compared with Switch, DTS has higher sample efficiency (i.e., requires less data samples) and yields a 2x speed-up to reach the same validation perplexity (Figure 3a), as well as higher FLOPs-efficiency and a 1.25x speed-up (Figure 3b). Experiments are conducted on OpenWebText datasets (40GB), with the GPT-MoE model architecture (24-layer MoE Transformer, 16 experts for every other layer, 1.5B parameters in total), which is the same parameter size and dataset as the largest GPT-2.
>
> **Q2: Why use FLOPs instead of actual training time as the metric?**
>
> Thanks for your concerns! Because the actual training time could be affected by the system engineering efforts on the implementation details, which are not our focus in this approach. Instead, in our experiments, we prefer to choose the computation complexity for fair comparisons. Detailed analysis (Appendix) shows that the routing pattern of DTS is relative stable, with heavy-loaded experts almost keep fixed. And we can utilize system techniques, e.g., partitioning hot experts among multiple GPUs and merging multiple cold experts onto less GPUs, for load balance between GPUs. Furthermore, we are also glad to explore the system-level optimization about how to deal with dynamic capacity of different experts in our future work.
>
> **Q3: Any masking or thresholding was used to evaluate DTS?**
>
> Thanks for pointing out this missing detail! We use threshold to drop the experts whose weights fall below the threshold as Figure 1, and no extra communication or computation will be wasted. In Appendix, we show the curve of average experts per token during the training iterations, where the FLOPs of MoE-layers decrease quickly at the early stage. We have already added this missing detail to the paper now.

---

> > ### Comment · Reviewer_gUQG · 2021-11-29
> > **Thanks for the response and additional details**
> >
> > **Q1: The end2end comparison experiment**
> >
> > Thanks for the clarification, figure 3 now better suports the efficiency claim (although I'm a bit confused that here you mention 1.42 speed-up, while Figure 3b shows 1.25).
> >
> > **Q2: Why use FLOPs instead of actual training time as the metric?**
> >
> > I understand the concern of the authors that measuring training time can bias results towards a particular implemenation, however measuring FLOPs alone has its own drawbacks, if those theoretical gains cannot be achieved in practice. I recommend reading the recent  draft "The Efficiency Misnomer" (https://arxiv.org/abs/2110.12894), which covers this topic broadly and has many interesting points why reporting  FLOPs (or runtime) _alone_ is not a good idea.
> >
> > **Q3: Any masking or thresholding was used to evaluate DTS?**
> >
> > Thanks for clarifying this in the paper.
> >
> > **Q4: Results about Validation set**
> >
> > Thanks for updating the figures.
> >
> > **Q5: The analysis about “effect of balanced loads” is not clear**
> >
> > Thanks for clarifying this as well.
> >
> > Given that you have addressed almost all my questions, I will update my review accordingly and increase my score. Again, the fact that the speed-ups are only measured using FLOPs and not real runtime on some hardware/implementation, is a bit unfortunate. In any case, I think that all the claims are better supported now (I would recommend emphasising in the paper that the 2x speed-up is regarding _data efficiency_, though).

---

> ### Author Response · Authors · 2021-11-23
> **Response to Reviewer 1. Part II**
>
> **Q4: Results about Validation set**
>
> Sorry for causing this concern. We have updated all the experiment figures with validation results through all the paper.
>
> **Q5: The analysis about “effect of balanced loads” is not clear**
>
> Thanks for pointing out this missing detail!
>
> First, we want to clarify that we record the average number of used experts per token during training and then calculate the FLOPs based on different iterations. Therefore, the x-axis of points in the figure of different plots are not equal.
> Meanwhile, we do not add the FLOPs for calculating balance loss throughout experiments, because we only want to compare the gate distribution with or without balance loss.
>
> *In addition, notice that one of the main practical reasons to use a balanced load (e.g. in Switch-MoE) is to efficiently use all available compute, which is not considered in the analysis presented*
>
> Current approaches in both NLP and Vision, both use a balance loss that not only works as balanced router, but also avoids the self-reinforcing imbalance, which uses only a few experts and leads to inefficient parameters [1][2].
> So, we analyze the influence of balance loss on DTS, to study whether it can also improve the model performance of DTS. And the results show that balance loss has little influence on the final model performance of DTS.
>
> *Figure 6a shows that the use of balanced loads adds an overhead of approximately 5⋅1011 FLOPs in the openwebtext dataset. Why? Why only in this task but not in the Wikipedia task?*
>
> The reason balanced loads add an overhead is that the distribution of gates becoming one-hot is slow down by the balance loss, and DTS-with-balance will involve more expert's computation. For example, the average used experts per token at the 1500 iteration is 5.58 for balance and 5.34 for imbalance.
>
> Reference:
> [1] Ruiz C R, Puigcerver J, Mustafa B, et al. Scaling Vision with Sparse Mixture of Experts. NeurIPS 2021.
>
> [2] Shazeer N, Mirhoseini A, Maziarz K, et al. Outrageously large neural networks: The sparsely-gated mixture-of-experts layer. ICLR2017

---

### Decision · Program_Chairs · 2022-01-20

**Decision:**

Reject

**Comment:**

This paper proposes a simple approach to improve the robustness of training a sparsely gated mixture-of-experts model, which at a high level simply consists in training initially as a dense gated model, to better warm start a final phase of sparse training. Results are presented to highlight the potential benefits of this approach.

The authors have provided a detailed response and updated results, in response to the reviews. Each reviewer has also responded at least once to the author response. Despite that engagement, all reviewers are leaning towards rejection (though there is one reviewer with a rating of 6, they regardless state that "I'm confident this will make a great resubmission at a future venue", indicating they actually support rejection).

The reviewers point out that the proposed method is not really novel, pointing to an existing recent paper. Even without that prior work, I would also argue that the proposed approach is conceptually straightforward and has benefits that were fairly predictable and not particularly surprising. Given the generally lukewarm reception from the reviewers, I think there is a legitimate concern to be had here about this work's potential for impact.

Though the review process has definitely improved the paper's manuscript since its submission, I unfortunately could not find a reason to dissent from the reviewers' consensus that this submission is not ready to be published. Therefore recommend it be rejected at this time.